# A Clinical and Pathophysiological Overview of Intestinal and Systemic Diseases Associated with Pancreatic Disorders: Causality or Casualty?

**DOI:** 10.3390/biomedicines11051393

**Published:** 2023-05-08

**Authors:** Maria Cristina Conti Bellocchi, Stefano Francesco Crinò, Giulia De Marchi, Nicolò De Pretis, Andrew Ofosu, Federico Caldart, Rachele Ciccocioppo, Luca Frulloni

**Affiliations:** 1Diagnostic and Interventional Endoscopy of Pancreas, Pancreas Institute, University of Verona, 37134 Verona, Italy; stefanofrancesco.crino@aovr.veneto.it; 2Gastroenterology Unit, Department of Medicine, Pancreas Institute, University of Verona, 37134 Verona, Italy; giulia.demarchi@aovr.veneto.it (G.D.M.); nicolo.depretis@univr.it (N.D.P.); federicocaldart94@gmail.com (F.C.); rachele.ciccocioppo@univr.it (R.C.);; 3Division of Gastroenterology and Hepatology, University of Cincinnati College of Medicine, Cincinnati, OH 45267, USA; andyofosu@gmail.com

**Keywords:** pancreatic disorders, inflammatory bowel disease, IgG4-related disease, autoimmune pancreatitis, exocrine pancreatic insufficiency, celiac disease

## Abstract

The relationship between chronic intestinal disease, including inflammatory bowel disease (IBD) and celiac disease (CelD), and pancreatic disorders has been little investigated. Although an increased risk of acute pancreatitis (AP), exocrine pancreatic insufficiency with or without chronic pancreatitis, and chronic asymptomatic pancreatic hyperenzymemia have been described in these patients, the pathogenetic link remains unclear. It may potentially involve drugs, altered microcirculation, gut permeability/motility with disruption of enteric-mediated hormone secretion, bacterial translocation, and activation of the gut-associated lymphoid tissue related to chronic inflammation. In addition, the risk of pancreatic cancer seems to be increased in both IBD and CelD patients with unknown pathogenesis. Finally, other systemic conditions (e.g., IgG4-related disease, sarcoidosis, vasculitides) might affect pancreatic gland and the intestinal tract with various clinical manifestations. This review includes the current understandings of this enigmatic association, reporting a clinical and pathophysiological overview about this topic.

## 1. Introduction

The co-occurrence of intestinal and pancreatic diseases and the common involvement of both organs in systemic diseases have been barely investigated. Postulating a similar mechanism in the pathophysiology of this association, the role of impaired gut homeostasis in influencing systemic inflammation might be crucial [1].

In this review, a comprehensive literature research has been performed including the most relevant studies up to December 2022 reporting the co-occurrence of pancreatic and gastrointestinal disease, including inflammatory bowel disease (IBD), celiac disease (CelD), and systemic inflammatory disease, to investigate the epidemiological, clinical, and pathophysiological features of these associations.

## 2. Pancreatic Disorders in Inflammatory Bowel Diseases

IBDs, including Crohn’s disease (CD) and ulcerative colitis (UC), are highly heterogeneous, chronic immune-mediated diseases affecting the gastrointestinal tract, with unknown but multifactorial etiology involving a complex interaction between the genetic, environmental, and/or microbial factors and the immune responses [2].

A wide spectrum of pancreatic abnormalities has been reported among patients with IBD, ranging from acute or chronic inflammation of the pancreas to asymptomatic imaging alterations or elevation of the pancreatic enzymes without any clinical relevance [3]. The occurrence of these conditions is often challenging for clinicians, due to the difficulty in finding an etiology, the decision to withdraw important drugs, and the management of the recurrences or the long-standing conditions. Acute pancreatitis (AP) is the most frequently observed pancreatic disorder in IBD patients; however, data about the diagnostic workup, clinical management, and outcome of these patients are lacking. The diagnosis of chronic pancreatitis (CP) with exocrine pancreatic insufficiency (EPI) might overlap with the symptoms of IBD and make it difficult to manage the patient. Chronic pancreatic enzyme elevation, moreover, is a benign but little-known condition. Its recognition and exclusion of pancreatic disease are essential to avoid clinical decisions that have an impact on the patient’s management.

### 2.1. Acute Pancreatitis

Acute pancreatitis (AP) consists of a sudden inflammation of the pancreas caused by the inappropriate activation of local digestive enzymes, with various grades of severity, diagnosed when at least two out of the following three criteria are present: (i) a typical pancreatic pain (high-intensity epigastric pain that can radiate to the upper quadrants and the back, with or without nausea and vomiting), (ii) a threefold or more elevation of the serum pancreatic amylase and/or lipase, and (iii) typical imaging findings [4]. The crude pooled incidence of AP is reported between 33 and 74 cases (95% CI 23–33; 48–81) per 100,000 person-years [5]. AP represents the most common pancreatic disorder in IBD patients with a twofold and fourfold risk in UC and CD, respectively, compared to the general population [6].

The most common etiologies of AP in IBD patients include gallstones and medications, while alcohol-induced AP seems less common in IBD compared to the general population. Less frequent etiologies include metabolic causes, autoimmune pancreatic involvement, and duodenal obstruction or papillary inflammation [3]. When AP occurs in IBD patients, both pancreatic and intestinal disease outcomes appear unaffected by one another [7,8].

Gallstones are frequently described in IBD patients, mainly in CD, involving up to 34% of cases [9]. Due to inflammation or previous ileal resection, impaired enterohepatic circulation leads to the hepatic excretion of cholesterol-supersaturated bile [10] and increased conjugated and unconjugated bilirubin excretion, with an unbalancing of the Admirand–Small triangle [11] and subsequent risk of biliary stones [12]. Additionally, prolonged fasting, parenteral nutrition use, use of non-steroidal anti-inflammatory drugs, and duration and activity of bowel disease could be contributing factors [13]. However, the diagnostic workup at AP onset in IBD patients, especially in the presence of gallstones, is often unclear and heterogeneous (Table 1).

According to universally recognized criteria, in the care of AP, examinations necessary for diagnosing gallstone-induced acute pancreatitis should include blood tests and ultrasonography (US). In the presence of increasing ALT levels over 150 U/L, the sensitivity ranges between 48 and 93% and the specificity ranges between 34 and 96%. Even more so, when an alteration by blood tests is detected in more than three of the items including bilirubin, ALP, GGT, ALT, and the ALT/AST ratio, 85% for sensitivity and 69% for specificity are reported. Finally, the combination of US and blood tests yields a sensitivity of 95–98% and a specificity of 100%. To date, several studies about AP in IBD patients only consider the presence of gallbladder stones to formulate a biliary-related AP diagnosis and often no data about the imaging study, and even more so, laboratory tests, are available [18,19].

Therefore, a prompt search for the etiology should be conducted when an AP episode occurs, and the gallstones should have priority both for their frequency in IBD patients, and for the need to choose the correct treatment policy, i.e., the cholecystectomy need, followed by the possible role of medications. Due to hypersensitivity or less frequent direct toxicity [14], drug-induced pancreatitis (DIP) requires caution and the presence of the following diagnostic criteria: (i) temporal sequence between medication introduction and the onset of AP; (ii) symptom cessation after drug discontinuation; (iii) AP recurrence after drug re-exposure [15]. According to the classification by Badalov et al. about the potential to induce AP based on the literature data, Azathioprine (AZA) and 6-mercaptopurine (6-MP), mesalamine, sulphasalazine, and metronidazole are classified as definite, while cyclosporine, prednisone, and prednisolone, are considered a possible cause of AP, albeit with less evidence [15].

In the last decade, AZA has been increasingly studied, while the role of mesalamine in inducing AP has been called into question [16]. In a Scandinavian nationwide register-based pediatric cohort, AZA was found to be associated with an increased risk of AP (incidence rate ratio 5.82 [95% CI 2.47–13.72]; events per 100 patients) during the 90-day risk period [20]. In adult patients, the incidence of AZA-induced pancreatitis seems to be higher, up to 7.3% in 510 patients treated with AZA, from a multicenter prospective registry. Several studies have shown the occurrence of DIP after a few weeks from AZA introduction, unrelated to dosage or other AZA-related adverse effects and typically affecting CD patients. A moderate clinical course, without hospitalization or a brief hospital stay, and a rapid recovery after drug withdrawal are described in most cases [14,21,22]. In a recent study, smoking, concurrent therapy with budesonide, and single-dose AZA were found as predictors of AZA-induced pancreatitis [23]. Interestingly, the AP occurrence within the first month by the initiation of AZA therapy is not similarly observed when thiopurines are used to manage other diseases. This association remains unclear but is probably related to a genetic predisposition. Polymorphisms in the gene encoding thiopurine methyltransferase enzyme are associated with dose-dependent adverse effects, including myelosuppression and hepatotoxicity, but are unrelated to AZA-AP risk. Recent studies identified the association of the HLA-DQA1*02:01-HLA-DRB1*07:01 haplotype with a 17% risk of developing pancreatitis in patients homozygous for the at-risk allele. Despite the paucity of data about the translation of these observations in clinical practice and the cost-effectiveness rate, a possible role of an immune-mediated mechanism in this association has been suggested [24].

In a relevant proportion of IBD patients, the etiology of AP remains unknown, and idiopathic AP (IAP) is eventually diagnosed, historically referred to as true extraintestinal manifestations, influenced by IBD activity with a rapid response when bowel disease is treated [3]. These data are not confirmed in a recent study on this topic, where 38 IAP (20.6%) among 185 IBD patients with an AP episode were retrospectively selected and studied [15]. No relationship was found between the extent/activity of disease or other extraintestinal manifestations, and a mild course was observed in all cases. Interestingly, additional diagnostic workup (magnetic resonance cholangiopancreatography, MRCP, and/or endoscopic ultrasound, EUS) after the first episode of IAP was performed in less than half of the patients, and an autoimmune pancreatitis (AIP) diagnosis was made in 13% of the cases of first IAP patients, due to a recurrence during the follow-up period. Thus, it is probable that a correct diagnostic route at the onset, with a more extensive workup including MRCP and/or EUS, could clarify the etiology of AP in IBD patients, reducing the percentage of IAP and preventing new AP episodes or pancreatic changes with an evolution toward chronic pancreatitis (CP) [25]. Moreover, in about 20% of IAP, the pancreatic inflammation preceded IBD diagnosis, mainly in the UC and pediatric population, suggesting that AP could be an early event in an undiagnosed IBD, particularly UC in young patients [15].

#### Pathogenesis of AP in IBD Patients

The innate immune activation and the acinar cell inflammatory signaling play a pivotal role in the pathogenesis of AP, with the need to balance pro-inflammatory cytokines/chemokines (Tumor necrosis factor, TNF-α; interleukin-1, IL-1) and anti-inflammatory or regulatory molecules (IL-10). Intestinal dysfunction and secondary inflammatory issues aggravate AP retroactively and are prodromes of systemic complications [26]. So, changes in the microcirculation, gut permeability/motility, bacterial translocation, and activation of the gut-associated lymphoid tissue, variously present in IBD patients, were postulated as factors promoting pancreatic inflammation.

In vitro studies aimed to prevent intestinal barrier disruption, with modulation of the immune cell activation, inflammatory cytokine production, and pancreatic inflammatory signaling, have been conducted with promising results, but to date, with scanty reproducibility in the clinical setting. An essential role in maintaining intestinal barrier integrity seems to be attributable to microbiota, which has been increasingly studied in pancreatic disease. Several studies support the existence of a gut microbiota–pancreas axis, where pancreatic juice and pancreatic diseases may alter the gut microenvironment and composition [27] and conversely, the gut microbiota may influence the occurrence of pancreatic diseases [28]. Nevertheless, little is known about translating these observations into clinical practice to prevent and treat pancreatic diseases. The administration of probiotics in severe AP was only attempted in Gou et al.’s metanalysis, but the significant heterogeneity among trials does not allow us to conclude their efficacy in real-life experience [29].

The oxidative stress and the lysosomal damage/dysfunction, generally linked to neurodegeneration and IBD in humans, may represent another important AP pathogenetic pathway. Using antioxidant substances [26,30] and targeting lysosomes for cell death induced by lysosomal membrane permeabilization are considered additional potential strategies in recent studies. Interestingly, Sudhakar et al. showed that loss of a carbohydrate-binding intracellular protein, galectin 9 (Gal-9), in mice causes unstable lysosomes in autophagy-active selectively in intestinal and pancreatic cells at the steady state, with compromising autophagy and consequent increased susceptibility to disease pathogenesis. Gal-9 preferentially targets intestinal Paneth and pancreatic acinar cells, promoting autophagic degradation to prevent cell death. So, the lysosome dysfunction, due to loss of Gal-9, could serve as a shared cell-intrinsic defect in the intestine and pancreas, supporting the pathophysiologic link between IBD and extraintestinal manifestations (EIMs), especially pancreatitis [31].

### 2.2. Autoimmune Pancreatitis

The AIP is a peculiar form of pancreatitis, characterized by an immune-mediated inflammation of the pancreatic gland, with a rapid response to steroid therapy. It is histologically classified into two types, type 1 and type 2, with different clinical courses, biochemical markers, histological findings, and clinical outcomes regarding relapse rate. Even in the absence of a histological diagnosis, the International Consensus Diagnostic Criteria (ICDC) may be applied to reach a clinical diagnosis. Moreover, when no specific type 1 or 2 criteria are present, the not-otherwise specified AIP (type NOS) could be defined [29]. Rare and underdiagnosed, the AIP prevalence in the general population is still not confirmed but estimated as 0.82:100,000 [30]. The association between IBD and AIP has been described in 5% to 40% of patients [32,33,34]. While in type 1 AIP, CD or UC can be sporadically found, instead in type 2 AIP, the diagnosis of IBD is crucial for the diagnosis, even if histology is not definitive, according to ICDC (Table 2).

Elevated serum IgG4 levels characterize type 1 AIP, possibly associated with extrapancreatic manifestations [41] and often clinically characterized by jaundice and/or pancreatic mass. Abdominal pain, weight loss, and diabetes are other possible presentations. Elderly male subjects are usually affected, and a tendency to relapse is observed after treatment discontinuation in a relevant percentage of cases (up to about 60%) [34]. A combination of parenchymal and ductal imaging, extrapancreatic disease, serum IgG4 concentrations (twofold upper normal values), pancreatic histology, and steroid responsiveness are needed for diagnosis. Imaging findings include diffuse or multi-focal enlargement of the pancreatic gland (sausage-like shape) and thin peripancreatic edematous rim, with altered signal intensity or enhancement pattern after contrast injection. However, a focal presentation, mimicking malignancy, could be observed, and histology becomes essential to avoid useless surgery. Type 2 AIP usually affects young patients without a predilection for sex. No serological biomarkers are available, and IgG4 levels are typically normal. On histology, granulocytic epithelial lesions (GEL) are typically characterized by intraluminal and intraepithelial neutrophils, often leading to the destruction and obliteration of the duct lumen. Both diffuse and focal involvement of the pancreatic gland can be present, and no other extrapancreatic disease location is described, except for IBD and mainly UC, that occurs in a relevant percentage of type 2 AIP patients (up to 80% in our experience in a cohort of AIP-selected patient) [35]. The clinical presentations include an incidental finding of focal lesion mimicking cancer, AP, not specific abdominal pain/weight loss, and increased pancreatic enzymes. The IBD may be previously or subsequently diagnosed, compared to AIP onset, but usually, a concomitant diagnosis is made within 2 years [35,37].

The clinical relevance of these data, reported by studies with a high number of patients despite the rarity of AIP, is that awareness is needed when an AIP occurs to reach an early IBD diagnosis with subsequent strict management. Conversely, when an idiopathic AP occurs, AIP should be suspected in IBD patients. Conflicting data about the relationship between bowel disease activity, pancreatic inflammation, and IBD outcomes in AIP patients are available. While Lorenzo et al. suggests a possible association of UC and AIP as a consequence of systemic inflammation and a higher colectomy rate in UC-AIP patients over the follow-up period, compared to only UC patients [37] in other studies, these data are not confirmed, suggesting a different pathogenetic pathway and the need for further prospective studies to understand the real impact of AIP on UC outcome [8,35,38,39]. Finally, a relevant proportion of patients may be initially classified in NOS AIP type and reclassified during the follow-up period. A study published on this topic reported a 17.4% reclassification rate of NOS AIP to type 2 for developing IBD during the follow-up period [42].

#### Pathogenesis of AIP in IBD Patients

The pathogenetic correlation between IBD and AIP is still unknown, but the immune-mediated mechanism may be postulated. Type 1 AIP has been defined as the prototype of IgG4-related disease (IgG4-RD), where the pivotal role of IgG4 could also involve the bowel. However, the gastrointestinal location of IgG4-RD is rare and histologically proven CD and UC without IgG4 infiltration are sporadically associated with type 1 AIP [43].

The role of chronic inflammation and lymphocyte recruitment by inflamed tissue, known as “lymphocyte homing”, described in the last decade, could be the underlying pathogenetic link. In most chronic inflammatory disorders, including IBD and AIP, the formation of tertiary lymphoid tissues properly occurs when the lymphocytes are recruited [44] with the subsequent morphological and functional change of postcapillary venules into endothelial venule-like (HEV-like) vessels. In the first phase of this process, a mucosal addressin cell adhesion molecule 1 (MAdCAM-1), which under normal conditions is expressed only in gut-associated lymphoid tissue (GALT), has been studied. In inflamed sites of IBD, HEV-like vessels expressing MAdCAM-1 have been extensively found [45]. In a study by Kobayashi et al., the UC-inflamed site MAdCAM-1 protein can be post-translationally glycosylated with MECA-79+ 6-sulfo sialyl Lewis X-capped carbohydrates, on immunoblotting, especially in the active phase [46]. The distribution of MECA-79+ HEV vessels and the less spread positive MAdCAM1 was also found in pancreatic cells of AIP patients and salivary glands in sclerosing sialoadenitis, suggesting a common immune-mediated mechanism [45].

Differently, in type 2 AIP, the frequency of IBD is higher and histological resemblances between the diseases have been described. The same neutrophil infiltration involving colonic crypt epithelium (cryptitis) and lumen (crypt abscess) in UC has been observed in the epithelium and lumen of pancreatic acini and small and medium-sized ducts in type 2 AIP [47]. The presence of shared antigenic molecules between both organs has been postulated, suggesting type 2 AIP as an extraintestinal manifestation of IBD, mainly UC.

### 2.3. Chronic Pancreatitis

Chronic pancreatitis (CP) is a multifactorial, fibroinflammatory syndrome characterized by repetitive episodes of pancreatic inflammation with subsequent infiltration of the pancreas by fibrosis, resulting in chronic pain, exocrine and endocrine pancreatic insufficiency, and reduced quality of life. Recurrent upper abdominal and back pain often appear as initial symptoms of CP, with attenuation as the disease stage advances and both exocrine and exocrine functions of the pancreas gradually decrease. Multiple complications, including chronic pain, characterize the end stage of CP, exocrine pancreatic insufficiency (EPI) and endocrine impairment, referred to as pancreatogenesis or type 3c diabetes mellitus, metabolic bone disease, and pancreatic ductal adenocarcinoma (PDAC) [48]. However, a painless pattern is described in about 10% of CP patients [49], even more than in IBD patients, where the disease is clinically unapparent in most cases. Occasionally, the diagnosis is suspected because of the onset of exocrine dysfunction [50,51].

The incidence of CP in the general population ranges from 5 to 12 per 100,000 with a prevalence of 50/100,000 person-years, and the risk seems to be increased in immune-mediated diseases, such as IBD [5]. The association of IBD with CP was postulated in 1950 when Ball et al. [52] described macroscopic and microscopic pancreatic alterations in autoptic specimens in up to 53% of UC patients. In a more recent series, a higher incidence for IBD patients has been described (5.75 vs. 0.56/10,000 person-years, respectively) compared to the general population [9,53]. In addition, a nationwide population-based cohort study revealed a 10.3-fold higher risk of IBD in patients with CP compared to a control group [53].

#### Pathogenesis of CP in IBD Patients

Few studies explored the association of CP in IBD patients, mainly before the 2000s, thus including cases of mass-forming CP, with a possibility to include the known AIP. Thus, although the presence of pancreatic duct changes in IBD patients, both on ERCP and EUS imaging studies [51], the pathogenesis and the possible outcome remain elusive. Genetic, immunologic, and obstructive causes could be involved, but no human studies are available to date.

In animal models of CP, when the inflammatory cytokine IL-1 is overexpressed under the control of the elastase promoter in the pancreas of mice, prominent histologic features of CP in terms of fibrosis and an inflammatory response dominated by T cells has been observed. Interestingly, despite the fibrotic replacement, the IL-1 transgenic mice developed neither pancreatic exocrine nor endocrine insufficiency after 8–10 months of life, in a comparable pattern of pancreatic alterations in IBD patients [54]. The role of IL-1 has also been recognized in animal models of bowel inflammation, as well as in cultural cells of IBD patients [55]. It is conceivable that in genetically predisposed IBD patients, the bowel disease activity with systemic cytokine- and, specifically, IL-1-mediated inflammation may involve other organs, as a pancreatic gland, without necessarily determine clinically evident CP.

Another cytokine of the IL-1 family, IL-33, has been proposed as the missing link between IBD and CP. The IL-33 produced by myofibroblasts and epithelial cells has been shown to increase the Th1 responses associated with CD, especially when the intestinal epithelium is damaged, as well as to mimic CU in experimental models with circulating levels correlating with bowel disease activity [56]. Moreover, it can promote the activation of key players in fibrogenesis occurring in intestinal and pancreatic inflammation. The enhanced concentration of IL-33 in acinar cells and pancreatic stellate cells and the IL-33-mediated production of profibrogenic molecules suggest its role in CP pathogenesis [57]. Further studies are needed to estimate the actual prevalence of CP in IBD patients and to evaluate the correlation of CP with IBD, mainly with bowel activity flares over the years.

Interestingly, a significant gut microbiota dysbiosis has been reported in CP patients, with increased opportunistic pathogens and depleted reduced taxa with a potentially beneficial role in intestinal barrier function. However, it is reasonable that these changes are primarily the result, rather than the cause, of the CP, probably due to the altered composition of the intestinal content, with resulting changes in the availability of substrates for microbial metabolism [27]. Moreover, no data about the effects of gut microbiota composition on the progression and severity of CP are available. So, the relationship between IBD and CP, mediated by microbiota alterations, cannot be proved.

Finally, considering the occurrence of pain in the setting of CP as well as in the natural history of IBD, the role of the enteric nervous system (ENS) and neuronal plasticity might be an interesting field for investigation [58].

The enteric neurons and intestinal immune cells share common regulatory mechanisms, responding to environmental factors, and the relay of neurotransmitters and neuropeptides is indispensable for effective immunity and tissue homeostasis. Emerging studies [59,60,61,62] have started to elucidate the role of specific immune mediators, including mast cells and the related cytokine release, macrophages and T cells, with or without the interaction with microbiota, in mediating pain sensitivity in the IBD context. The translation of these observations in the clinical setting have led to the possibility to stimulate the vagus nerve for the pain control and outcome in CD patients with promising results [63].

Similarly, the role of ENS and intrapancreatic ganglia might be involved in the pathogenesis of pancreatic pain, specifically in CP, due to neural morphologic changes triggered by chronic inflammation and microenvironment alteration [64].

### 2.4. Exocrine Pancreatic Insufficiency

EPI consists of a deficiency of pancreatic enzymes, resulting in the inability to digest food properly. It is usually associated with pancreatic diseases, mainly CP, due to the progressive depletion of pancreatic exocrine function. However, the cause may be the alteration in the digestive chain involving the pancreas, including pancreatic stimulation, pancreatic juice production or outflow, and synchronization with gastrointestinal secretions for mechanical and functional reasons [65]. Thus, pancreatic and extrapancreatic diseases such as IBD may be associated with EPI.

Symptoms of EPI include steatorrhea, abdominal distension, flatulence, and/or weight loss, whose intensity may range from scarce symptoms to the impairment of quality of life. Moreover, the resulting malnutrition may be responsible for osteopenia, sarcopenia, reduced immunocompetence, and a nutritional deficiency [65].

The prevalence of EPI in the general population is unknown. It is mostly a late-stage manifestation of CP, occurring when 90% of the pancreatic gland is lost, but it could be influenced and anticipated when recurrent pancreatic disease flares occur [66]. On the other hand, the frequency of EPI in IBD patients varies considerably, ranging between 18 and 80% of cases, depending on patient selection and the diagnostic tests used (PABA test, secretin-caerulein test, qualitative or quantitative fecal fat test, and more recently fecal elastase-1, FE-1). In a prospective sectional study by Maconi et al., 14% of patients with CD and 22% of patients with UC suffered from EPI, using the FE-1 levels as a diagnostic tool. Compared to control subjects, the odds ratios (OR) for EPI were 8.34 for patients with CD and 12.95 for patients with UC. Moreover, in CD, EPI seems to be related to the extension and activity of bowel disease, mainly in the ileal location [67].

If, on the one hand, the symptoms of IBD may overlap with EPI, making it underdiagnosed, it should be noted that tests such as FE-1 are strictly influenced by the loose/watery stool typical of patients with active IBD. The administration of pancreatic enzyme replacement therapy (PERT), in association with dietary counseling and vitamin supplementation, demonstrated high effectiveness in ameliorating symptoms and improving quality of life. However, no significant studies and/or guidelines address the possible effects of PERT in IBD patients.

#### Pathogenesis of EPI in IBD

Some authors suggest that EPI in IBD patients may result from idiopathic CP, secondary to ductal obstructive changes, or previously undiagnosed AIP. However, as mentioned above, EPI in CP is a late-stage symptom, and the entity of ductal changes is insufficient to explain the EPI onset in these patients. Another hypothesis of IBD and EPI association is the immune-mediated mechanism, suggested by the presence of autoantibodies against exocrine pancreas (PABs), found in up to 39% of CD patients and up to 23% of UC patients. The PABs belong to the IgG and IgA isotypes and show two distinct response patterns, suggesting at least two different pancreatic autoantigens as targets of the autoimmune responses. The identity of PAB-specific autoantigens has been recently elucidated as a glycoprotein predominantly expressed in the pancreas and known as GP2. It was previously believed that GP2 was exclusively expressed by pancreatic acinar cells, but recent studies showed it also to be in the epithelium of Peyer’s patches. However, the role of exocrine pancreatic autoantibodies in inducing EPI was postulated in other autoimmune diseases, such as by Sjogren [58]; no data about the association of PABs with EPI or with morphologic changes in the pancreatic duct in IBD patients are available. In a study by Barthet et al. assessing the frequency of radiological and biological alterations in IBD patients with a previous AP episode by comparing data with IBD patients without a history of AP, no differences were found in serum levels of PABs (*p* = 0.17), although the higher rate of reduced FE-1 in AP experienced patients [68]. To date, the role of PABs has been studied as a possible disease marker in association with ANCA and ASCA autoantibodies rather than as a cause of EPI. Only one study [69] documented the most frequent presence of extraintestinal autoimmune manifestations in CD patients with PAB compared to seronegative cases. Further prospective studies are needed to explore the clinical meaning of these observations.

### 2.5. Chronic Asymptomatic Pancreatic Hyperenzymemia (CAPH)

Pancreatic enzyme elevation is a relatively common occasional finding during laboratory examination. Any elevation of amylase and/or lipase above the upper normal value confirmed on at least two occasions spaced over time is defined as chronic pancreatic enzyme elevation (CAPH). Even if pancreatic enzymes are the preferred serological test for the diagnosis of pancreatitis, its detection may not be associated with pancreatic disease and related to extrapancreatic causes such as other isoenzyme production (for example, amylase by salivary glands), the reduced clearance for renal impairment or macrolipase formation, other hepatobiliary/gastrointestinal diseases with reabsorption through an abnormally permeable intestinal mucosa, neoplasms with paraneoplastic production, as well as diabetes, drugs, and infections [70].

The exact prevalence of CAPH in the general population is still unknown. While an Italian study reported a 2% prevalence in a large sample of the general population, among 4964 subjects, who had undergone pancreatic enzymes blood testing [71], in a German study, CAPH was found to occur in 8% of hospitalized patients without pancreatic diseases [72]. Additionally, hyperenzymemia may be found in a relevant proportion of diabetic patients. Baseline data from over 9000 subjects in the LEADER trial, a study about the cardiovascular safety of Liraglutide in type 2 diabetes, documented elevated lipase and/or amylase in nearly 25% of cases [73].

In IBD patients, CAPH seems more frequent and is found in 8% to 21% of cases, with values higher in CD than in the patients with UC and severe and moderate clinical activity than in the patients with mild and inactive disease. Moreover, some authors found higher median serologic and urinary isoamylase values in smoking patients [74,75].

In all cases, a hyperenzymemia should be investigated through imaging studies, possibly finding undiagnosed neoplasms, CP, pancreatic cysts, or benign pancreatic condition as AIP in 2.2%, 16.2%, 12.8%, and 17.2%, respectively, as reported in a recent metanalysis [74]. The confirmed CAPH, in the absence of pancreatic alterations, has been described and followed over a long follow-up period, with no pathological meaning being documented [75].

#### Pathogenesis of CAPH in IBD Patients

Several pathogenetic mechanisms were suggested to explain CAPH in IBD patients, mainly the pancreatic release of enzymes as the effect of inflammatory mediators from the inflamed bowel, as observed in other intestinal diseases. However, the observed hyperenzymemia in inactive IBD may suggest other potential mechanisms, such as immunological ones. This evidence is supported by the relationship between CAPH and both AIP (up to 18% of cases) and primary sclerosing cholangitis (CSP), where imaging reveals the presence of pancreatic ductal changes in about 50% of cases [35,76]. In these selected patients, a combination of pancreatic outflow obstruction and immunological factors may be present. It is worth mentioning that this is due to CAPH associated with drugs used in IBD management, mainly Azathioprine, due to the risk of withdrawal from important therapy. Except for cases of DIP, which, as mentioned above, usually occurs during the first weeks of treatment and requires specific criteria, it is not likely that the elevation of pancreatic enzymes, not associated with typical pain or imaging alterations, is caused by medical therapy [76,77]. Moreover, the lipase increases above the upper limit of normal after AZA induction is rare and not predictive of the development or severity of AZA-induced AP [21]. In case of pancreatic enzymes increasing, a second level of imaging should be performed to promptly recognize the small percentage of patients with pancreatic alterations (AIP, or pancreatic lesions), adopting the right management. On the contrary, no additional examinations would be performed in case of negative results. The therapy should be continued since it is probable that hyperenzymemia also persists after drug discontinuation due to the absence of causality [78].

### 2.6. Pancreatic Cancer in IBD Patients

Traditionally, IBD patients are considered to have an increased risk of intestinal and extraintestinal cancer compared to the general population [79]. Regarding pancreatic cancer, several studies have found a risk of up to 14-fold in CSP, associated with IBD in most cases, compared to the general population [80]. However, when IBD-alone patients are considered, conflicting results are reported [80,81,82]. In a recently published binational population-based study, the adjusted hazard risk for pancreatic cancer in IBD patients compared to the general population was 1.43 (1.30–1.58), slightly higher for CD than UC and for men than women. Moreover, when a CSP diagnosis was present, the overall hazard risk increased to 7.55 (4.94–11.5) but with a decrease over the years (5–10 years) from biliary disease onset, suggesting a detection bias or a misclassification of distal cholangiocarcinoma [83]. Moreover, the cumulative incidence was lower than 1% after at least 20 years, and this cancer risk seemed to be reasonably reduced [83,84]. Different reasons linked to pancreatic disorders in IBD patients, use of immunomodulators and immunosuppressant therapies, and the long-standing systemic chronic inflammation have been suggested to explain the increased risk of pancreatic cancer in IBD patients, without clear evidence.

## 3. Pancreatic Disorders in Celiac Disease

Celiac disease (CelD) is an immune-mediated enteropathy triggered by dietary gluten in genetically predisposed subjects, responsible for a chronic inflammatory process in the small intestine with consequent malabsorption and/or extraintestinal manifestations. Previously considered a rare disease, prevalent in childhood, with classic manifestations of the malabsorption syndrome, CelD epidemiology has changed radically in the last decades [85]. Both endocrine and exocrine pancreatic impairment may be caused by or co-exist with CelD.

The pancreas–gut axis suggested to explain the connection between type 1 diabetes (T1D) and CelD, as well as the role of gluten and intestinal permeability in altering glucose homeostasis and insulin sensitivity, are mentioned in the relationship of pancreatic endocrine dysfunction and CelD [86]. On the other hand, exocrine involvement includes AP, EPI, and CAPH, with an interesting impact on clinical management.

### 3.1. Acute Pancreatitis in CelD Patients

The prevalence of CelD in the general population of Western countries is about 1%, with a male/female ratio of 1:2.5, and the diagnosis is increasingly made in adulthood [85]. Almost a threefold increase in the risk of AP has been described in CelD patients in population-based design studies [85,87,88]. No higher risk of gallstones has been found, and usually, AP etiology remains unclear, more frequently involving the female sex and at a younger age, with smoking detected as a risk factor. Moreover, while one retrospective large US cohort study describes less severe forms, with lower morbidity and mortality [89], another large population-based study documented a worse outcome with a higher risk of developing early and delayed AP complications [87].

Several factors are believed to contribute to the pathophysiology of pancreatitis in CelD, including malnutrition, intestinal inflammation with disruption of enteric-mediated hormone secretion (cholecystokinin and secretin), or papillary involvement [90]. Moreover, although few cases of AIP associated with CelD have been described, the role of autoantibodies, extensively found in both diseases, and the lack of information about the pathogenetic link between pancreatic disorders and CelD led to postulate a possible shared immunological trait. However, to date, a low probability of association was found [91].

### 3.2. Exocrine Pancreatic Insufficiency in CelD Patients

The prevalence of EPI ranges from 4% to 80% in patients with untreated celiac disease [92]. Usually, EPI in CelD may be unrelated to structural changes in the pancreatic parenchyma and might be reversible by a GFD in most patients, probably because of the direct effect of duodenal villous atrophy or overdiagnosis related to FE-1 dilution in watery stools [93]. However, several studies have shown the effectiveness of PERT in improving the frequency of stool over time in association with increasing levels of FE-1 [94] and body weight gain, especially in pediatric patients [95].

The pathophysiological mechanisms of EPI in CelD patients may be multifactorial but primarily related to the impaired secretion of pancreatic stimulating hormones from the duodenum, resulting in an inadequate postprandial response to intraluminal content. The mucosa atrophy is responsible for altered synthesis, storage, and/or secretion of secretin and CCK [96], with subsequent reduced pancreatic stimulation and secretion, asynchrony between gastric emptying and gallbladder contraction, and fat maldigestion [92]. Moreover, malnutrition and the small peptides’ malabsorption reduced the production of pancreatic enzyme precursors [97], and structural pancreatic changes sometimes observed in CelD patients may be due to other contributing mechanisms [98]. Due to the observed autoantibodies production in both AIP and CelD and the serological and tissue IgG4 increase in sporadic cases, Leeds and Sanders [99] postulated a role of the IgG level, particularly IgG4 fractions, as the missing link between EPI and CelD, but data are lacking.

### 3.3. CAPH in CelD Patients

CAPH has been described in CelD patients because of subclinical pancreatic inflammation, reduced hormonal input due to duodenal atrophy, or immunocomplex formation with subsequent macroamylasemia. In a study by Carroccio et al., a considerably high frequency (about 25%) of CAPH in childhood and adult CD was found, suggesting the role of elevated serum pancreatic enzymes in suspecting CD during screening laboratory tests, similar to hypertransaminasemia [100].

Conversely, the search for CelD in 65 CAPH patients was essentially negative compared to an Italian study’s 1550 healthy control group [101]. Therefore, further studies are needed to suggest the extensive dosage of pancreatic enzymes and the serological CelD investigations in case of unexplained CAPH.

### 3.4. Pancreatic Cancer in CelD Patients

In the last decade, several studies have connected the presence of CelD with the risk of malignancy onset. Despite the well-known risk of T-cell lymphoma and small bowel adenocarcinoma, the risk of other organ malignancies, including pancreatic cancer, is still uncertain [102]. A published study on this topic, mainly conducted in northern European countries, concluded a slightly elevated risk of pancreatic cancer, ranging from 1.2 to 2.1 [103,104] (Table 3).

For other malignancies, the risk seems to be primarily restricted to the first year after CelD diagnosis, probably as an effect of the protective role of GFD in the subsequent years [107]. This advantage does not seem applicable for pancreatic cancer, whose risk of 1.6 (1.32–2.10) also persists after the first year, mainly in men and those of older age [103]. However, a real correlation between CelD and pancreatic cancer cannot be established, and further studies about this association should be conducted for real risk and pathogenetic links.

## 4. Systemic Diseases with Both Gastrointestinal and Pancreatic Involvement

Several systemic diseases may present the coexistence of pancreatic and gastrointestinal involvement, and knowing this association may be helpful for clinicians in the patient’s management. The most frequent systemic inflammatory and vasculitic diseases with this occurrence are described below.

### 4.1. IgG4-Related Disease

Immunoglobulin G4-related disease (IgG4-RD) is a multi-organ immune-mediated condition characterized by a tendency to form tumefactive lesions, a dense lymphoplasmacytic infiltrate rich in IgG4-positive plasma cells, storiform fibrosis, frequent elevations of serum IgG4, and a rapid response to steroids [108]. The more common disease locations are the pancreas, bile ducts, salivary and lacrimal glands, kidney, and retroperitoneum, but virtually all organ systems may be involved. To facilitate the early recognition of this condition, four distinct clinical phenotypes have been described with different percentages of specific disease location: the pancreato-hepato-biliary disease group; the retroperitoneal/aortic involvement group; the head and neck-limited disease group; Mikulicz syndrome with systemic involvement [109].

Type 1 AIP is regarded as a prototypical organ manifestation of IgG4-RD (especially not exclusively group one), which can occur alone or simultaneously or metachronously with other organ involvement, as described above.

Conversely, gastrointestinal involvement is rare and heterogeneous, reported by a single clinical observation and small series [110]. Sporadic cases of gastric [111,112] duodenal [113] and small bowel [114] mass or ulcerative lesion-mimicking cancer have been reported, with a postoperative diagnosis of IgG4-related lesions. Clinically mass-related symptoms with gastric outlet obstruction or bleeding by ulceration could present. No detailed criteria are available for diagnosing IgG4-related disease restricted to the gastrointestinal tract. The association with other systemic manifestations, elevated serum IgG4 levels (≥135 mg/dL), and abundant lymphoplasmacytic infiltration rich in IgG4+ cells/HPF fibrosis (>10 and IgG4+/IgG+ ratio of >40%) on histopathological examination was suggested in 2012 [115]. Since other inflammatory conditions that involve the upper GI tract, such as atrophic gastritis, IBD, CelD, hematologic disease, and polypoid syndrome, may also reveal the same phenomenon, the increase in IgG4-positive cells alone is not specific to gastrointestinal IgG4-RD and requires caution [116,117,118]. In relation to the tendency to undergo surgery, no data about treatment and response to steroid treatment are available in these patients, except for gastric involvement, which may respond to antisecretory drugs [110]

Similarly, colonic involvement in IgG4-RD is infrequent, with a clinical course similar to IBD, but diagnosis is potentially difficult since a high risk of overlap may be present. Suppose the colonic infiltration of IgG4 has been described in the bowel samples of type 1 AIP patients, suggesting an extrapancreatic location of IgG4-RD [40,43] on the other hand. In that case, IgG4+ plasma cells are frequently increased in the colonic specimen of IBD patients. In the study by Rebours et al., the IgG4+ plasma cells in the gut mucosa reflect more on the inflammatory context than specific AIP disorders, probably related to dysregulation of the immune response [119]. Similarly, Kuwata et al. [116] postulated the relationship between IgG4 bowel infiltration and disease severity, with loss of response to therapy and the need for surgery.

### 4.2. Sarcoidosis

Sarcoidosis is a chronic multisystemic granulomatous disease of unknown origin, which usually affects the lungs, lymph nodes, skin, and eye. Still, virtually all organs may be involved, with challenging diagnoses and dramatic responses to steroids [120]. The gastrointestinal involvement of systemic sarcoidosis is infrequent and usually located in the liver. Due to the rarity of this condition, only case reports are available about pancreatic location, which seems to involve about 1–5% in autoptic dated studies on sarcoidosis patients. The clinical presentation is that of an AP or a pancreatic mass, mimicking cancer, due to the infiltration of the pancreas. Although a diagnosis may be suspected in yet-diagnosed systemic sarcoidosis patients, using cross-sectional imaging and even more EUS [121,122], usually a post-surgical diagnosis is made, especially when a systemic diagnosis is lacking, so no extensive data about conservative management and steroid response of pancreatic location are available.

The involvement of the luminal gastrointestinal tract is rare for sarcoidosis, located at the gastric antrum or, even more rarely, gut wall. Symptoms may include abdominal pain, nausea, and early satiety, or sub occlusive symptoms with progressive weight loss [123]. Steroid therapy may not be enough to treat intestinal wall thickening, and surgery is generally needed.

### 4.3. Vasculitides

The vasculitides include several conditions characterized by inflammation of blood vessels, subsequent wall thickness, reduced hematic flow, and organ damage. Both gastrointestinal and pancreatic involvement may be caused by vascular inflammation in the following vasculitides: the Henoch–Schönlein Purpura (HSP), antineutrophilic cytoplasmic antibodies (ANCA)-associated vasculitis (AAV), polyarteritis nodosa (PAN), and rarely, secondary vasculitis caused by systemic lupus erythematosus (SLE) [124].

#### 4.3.1. Henoch-Schönlein Purpura

Also known as IgA-mediated leukocytoclastic vasculitis, HSP usually occurs in the pediatric population but can present at any age, triggered by infections, vaccinations, or drugs. The clinical manifestations are thought to arise from IgA depositions in blood vessel walls in the affected organs, mostly skin, with acute onset of palpable purpura mainly located on the lower extremities, joints, and, importantly, kidneys, whose involvement strongly influences the prognosis. Moreover, the gastrointestinal tract is involved in 50–75% of patients, with a predilection for the duodenum and descending colon, even if the ileal location and esophageal ulceration have been sporadically described. Symptoms include abdominal pain, melena, or diarrhea with hematic stools. Possible complications, especially when the small intestine is involved, include wall thickening, infarcts, necrosis, perforation, or intussusception, requiring surgery in up to 10% of cases [125].

Pancreatic HSP involvement is a rare but well-recognized cause of acute and recurrent pancreatitis, which can occur before the characteristic rash and present as the initial manifestation of HSP. Concomitant cholecystic and intestinal inflammation may be present, and as for the systemic disease, steroids seem to improve the clinical outcome in these patients [126,127].

#### 4.3.2. ANCA-Associated Vasculitis (AAV)

AAVs are rare idiopathic multisystem autoimmune diseases, more common in older people and men, characterized by the necrotizing inflammation of blood vessels that include granulomatosis with polyangiitis (GPA, previously known as Wegener’s granulomatosis), microscopic polyangiitis (MPA), and eosinophilic granulomatosis with polyangiitis (EGPA, previously known as Churg–Strauss syndrome) [128]. Although gastrointestinal involvement may be potentially observed in all AAVs, the pancreatic location of granulomatous inflammation has been described in GPA.

Typically, GPA inflammation involves small- and medium-sized blood vessels of the nose, upper respiratory tract, and kidneys. However, approximately 5% to 11% of GPA patients have an abdominal manifestation overall [129]. Though uncommon, gastrointestinal involvement can cause severe morbidity with perforations, bleeding, and even mortality. Esophageal, gastric, or intestinal ulcerations resembling IBD or ischemic findings may be found. Pancreatic involvement includes idiopathic AP, with subsequent diagnostic ANCA dosage and/or pancreatic or other organ histopathology, and granulomatous lesions mimicking cancer, diagnosed by histopathology after surgical intervention [130].

#### 4.3.3. Polyarteritis Nodosa (PAN)

PAN is characterized by a necrotizing, focal segmental inflammation of medium-sized muscular arteries involving kidneys and other organs (testicle, nervous system, skin), leading to a weakness of the arterial wall with subsequent stenosis, alternating to aneurysmal dilation and rupture (“rosary sign”).

Sometimes associated with infection by the hepatitis B or hepatitis C virus, and more rarely to Parvovirus B19 or Cytomegalovirus, PAN is usually diagnosed in the fourth–fifth decade, though pediatric cases have also been described. The gastrointestinal involvement ranges between 14 and 65% and usually is located in the gallbladder and small bowel, even if the colonic and appendiceal location has been described. The most common symptom is post-prandial abdominal pain related to impaired hematic mesenteric circulation resulting in focal and, in complicated cases, transmural ischemia [131]. Although the frequency of AP in PAN patients is reported to be as low as 3%, up to 37% of PAN autopsy cases have shown vascular changes involving the pancreatic gland associated with the systemic disease. Pancreatic involvement includes AP, pancreatic infarcts/pseudocysts (with or without intracystic hemorrhage), or masses [131,132,133]. Because of its rarity and lack of reliable discrimination from pancreatic cancer and the risk of necrotizing AP with severe complications when AP occurs, awareness about pancreatic involvement in PAN patients is important to reach an early diagnosis followed by prompt immunosuppressive treatment.

#### 4.3.4. Systemic Vasculitis Associated with SLE

Systemic lupus erythematosus is a multisystemic autoimmune disease, virtually able to involve every organ system. Although abdominal pain is a common clinical presentation of SLE, the pathogenetic mechanism of both gastrointestinal and pancreatic involvement seems to be related to vasculitic changes secondary to immune complex deposition and thrombosis [134]. Also known as Lupus Enteritis, mesenteric vasculitis is an uncommon but well-recognized complication of SLE and clinically occurs as ischemic small bowel enteritis or colonic ulcerations. The symptoms vary from mild, nonspecific abdominal pain and bloating or loose stool to necrosis and intestinal perforation, manifesting as severe, extensive gastrointestinal bleeding or acute surgical abdomen. In cases where a rapid response to immunosuppressive therapy is not achieved, surgical intervention is needed. In addition, a protein-losing enteropathy in SLE has been described, probably due to the intravascular activation and conversion of complement, non-necrotizing vasculitis, acquired lymphangiectasia, and increased microvascular/endothelial permeability [135].

On the other hand, the rate of pancreatic diseases is much lower than in the gastrointestinal tract and does not reach 5%, according to published series [136]. The SLE-associated AP is difficult to diagnose as it is generally considered a diagnosis of exclusion. The vascular damage has been postulated as a pathogenetic cause of AP in SLE patients due to the thrombosis of arteries and arterioles, intimal thickening/proliferation, and immune complex deposition with complement activation in the wall pancreatic arteries [134]. Despite its rarity and difficulty in determining AP outcomes in these selected patients, the AP in LES children, who have fewer potential alternative AP causes than adult patients, showed a higher complication rate and mortality due to the macrophage activation syndrome. Moreover, a large cohort study described a higher risk for longer hospital stays, severe complications, and inpatient mortality in AP patients affected by SLE compared to AP patients without SLE, regardless of the etiology [135,136].

## 5. Conclusions

The co-occurrence of gastrointestinal and pancreatic disease may be challenging for clinicians, and doubts about the causality or casualty of this association may occur. This clinical and pathophysiological overview of pancreatic disorders in different gastrointestinal and systemic disease settings provides interesting insights about the possible role of systemic inflammation and the hyperactivation of the immune system. Further studies are needed to clarify the missed link in this association.

## Figures and Tables

**Table 1 biomedicines-11-01393-t001:** Studies investigating the frequency of acute pancreatitis in IBD patients.

Authors	AP Frequency	Number of Patients	Etiology	Recurrence Rate of AP	Diagnostic Criteria for AP
Bermejo, 2008 [14]	1.6%	82 (79% CD; 21% UC)	Drug induced 63% Gallstones 12.2% Idiopathic 20.7%	13%	Not specified
Garcia, 2020 [15]	1.5%	185 (68.7% CD; 31.3% UC)	Drug induced 59% Gallstones 18.4% Idiopathic 20.6%	5%	Not specified
Moolsintong, 2005 [16]	n/a	48 (50% CD, 50% UC)	Drug induced 17% Gallstones 21% Idiopathic 4%	20%	CT or US in 56% of cases and pancreatic abnormality found in 54% of them; pancreatic enzyme elevation
Weber, 1993 [17]	1.4%	12 (100% CD)	Not mentioned	16%	CT performed in 75% of cases and pancreatic abnormality in 77% of them

**Table 2 biomedicines-11-01393-t002:** Autoimmune pancreatitis in IBD patients.

Authors	Study Design	Total Number of Patients	Number of IBD-AIP Patients	Type of AIP According to ICDC	Findings about the Association
Conti Bellocchi, 2022 [35]	Retrospective; AIP cohort	267 AIP patients	45 (UC)	42 probable type 2 3 definitive type 2	69% simultaneous diagnosis or within 1 year; Mild extensive colitis Relapse rate 11%
Oh, 2019 [36]	Retrospective: AIP cohort	244 AIP patients	12 (UC)	12 probable type 2	33% simultaneous diagnosis
Lorenzo, 2018 [37]	Retrospective: IBD cohort	unknown	91 (58 UC, 33 CD)	12 definitive type 2 77 probable type 2 2 definitive type 1	At AIP diagnosis, 72% of patients had active IBD 26% simultaneous diagnosis, 24% 2 years later. Proctitis or extensive colitis Relapse rate 34% Increased colectomy rate compared to IBD alone patients
Kim, 2017 [8]	Retrospective: IBD cohort	3307 IBD patients	13 (UC)	4 definitive type 2 9 probable type 2	23% simultaneous All steroids treated, no recurrence
Ramos, 2016 [6]	Retrospective (IBD cohort)	Unknown	5 (UC)	3 probable type 2 1 probable type 1 1 definitive type 1	1 simultaneous and 4 after IBD (mean 9 months) 2 patients had PSC
Ueki, 2015 [38]	Retrospective (IBD cohort)	1751 IBD patients	7 (5 UC, 2 CD) [3 probable excluded]	7 definitive type 2	80% active IBD at AIP onset
Park, 2013 [39]	Retrospective (AIP cohort)	104 AIP patients	6 UC	4 type 2 2 unknown (based on histology)	100% extensive colitis
Ravi, 2009 [40]	Retrospective (AIP cohort)	71 AIP patients	4 (3 UC, 1 CD)	Type 1	100% extensive colitis

**Table 3 biomedicines-11-01393-t003:** Association between celiac disease and pancreatic cancer risk in population-based studies.

Authors	Total Number of Patients	Number of Pancreatic Cancers	Calculated Risk of Pancreatic Cancer in CelD Patients	Findings about the Association
Askling, 2002; Sweden [104]	11,019	9	1.9 (95% CI 0.9–3.6)	Adults CelD patients (but not children and adolescents) have a risk for pancreatic cancer that declines with time and eventually reaches unity.
Elfstrom, 2012; Sweden [105]	28,892	64	1.4 (95% CI 0.9–2.0)	Risk increase in the first year from diagnosis for all CelD, latent CelD and duodenal inflammatory findings.
Ilus, 2014; Finland [106]	11,991	45	0.7 (95% CI 0.5–0.9)	No increased risk of cancer in the whole series, but risk increased after 5 years from the diagnosis of celiac disease
Lebwohl, 2022; Sweden [103]	47,241	152	2.3 (95% CI 1.8–2.8)	Risk increase, confined to diagnosis > 40 yo, primarily present within the first year of diagnosis.

## Data Availability

No new data were created or analyzed in this study. Data sharing is not applicable to this article.

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
