# Peer review of "A Clinical and Pathophysiological Overview of Intestinal and Systemic Diseases Associated with Pancreatic Disorders: Causality or Casualty?"

_biomedicines, 2023, doi:10.3390/biomedicines11051393_

Round 1

Reviewer 1 Report

The authors in this review discuss the current knowledge on the pathogenesis of IBD and CelD in relation to pancreatic disorders.

However, before publication some points need to be clarified.

My comments:

General remark – please correct the position of reference numbers which should appear at the part of the sentence (now they are after the coma).

Line 34 - please add brief methodology of this review

Line 64 – please explain what NSAIDs stands for

Table 1 – references to authors listed in table 1 should be in order (so, Garcia 2020 should be [13]; Bermejo 2008 should be [14] etc.).

Line 237 – From histological point of view there are only four kinds of tissues: epithelial, connective, muscular and nervous. Therefore, such terms as “lymphoid tissue” (Line 21 and throughout the text) “fibrotic tissues” (line 42), pancreatic tissue (line 543), are not justified. The authors terribly confuse organs with tissues. Organs are assembled from the four basic types of tissues and have cells with specialized functions.

Like 281 and the rest of the text – the authors should ensure that the use of term “expression” is in relation to genes only.

Line 311 – please use refences in brackets.

Line 572 – change to the “small intestine”

Line 493 - From the clinical point of view one of the most important symptoms related to both intestine and pancreas disorders is pain. Pathogenesis of pain is associated with changes in both peripheral and central nervous system. Noteworthy, intrapancreatic ganglia and enteric ganglia have the same sets of neurotransmitters/neuromodulators and some authors believe that intrapancreatic ganglia are prolongation of the enteric nervous system. In my opinion the innervation of the intestine and pancreas is in line with current review and should be (at least briefly) presented/discussed.

Author Response

Editor-in-Chief

Ms. Alexandra Negura

Assistant Editor

Biomedicines

Dear Editors,

we are grateful for the opportunity of sending a revised version of the manuscript entitled “A clinical and pathophysiological overview of intestinal and systemic diseases associated with pancreatic disorders: Causality or casualty?” for your consideration for publication in Biomedicines. We also thank the reviewers for their insightful comments that improved the quality of our paper. 

Reviewer 1

The authors in this review discuss the current knowledge on the pathogenesis of IBD and CelD in relation to pancreatic disorders.

However, before publication some points need to be clarified.

My comments:

General remark – please correct the position of reference numbers which should appear at the part of the sentence (now they are after the coma). 

RE: The position of reference numbers has been checked and corrected.

Line 34 - please add brief methodology of this review

RE: Thank you for your comment. The following sentence has been included in the introduction: “A comprehensive literature research has been performed including the most relevant studies up to December 2022 reporting … “

Line 64 – please explain what NSAIDs stands for

RE: NSAIDs stands for “non-steroidal anti-inflammatory drugs”. We have spelled out “NSAIDs” in the text.

Table 1 – references to authors listed in table 1 should be in order (so, Garcia 2020 should be [13]; Bermejo 2008 should be [14] etc.).

RE: the correct sequence of references has been checked and corrected, thank you.

Line 237 – From histological point of view there are only four kinds of tissues: epithelial, connective, muscular and nervous. Therefore, such terms as “lymphoid tissue” (Line 21 and throughout the text) “fibrotic tissues” (line 42), pancreatic tissue (line 543), are not justified. The authors terribly confuse organs with tissues. Organs are assembled from the four basic types of tissues and have cells with specialized functions.

RE: Thank you for your clarification. However, the terms were used in specific sentences with different meanings:

-the mucosa associated (or gut-associated as reported in the text) lymphoid tissue is a recognized entity involved in the intestinal immune responses and the term is correctly used in the text.

-the term “fibrotic tissue replacement” was considered as the substitution/infiltration of normal tissues of the pancreatic gland by fibrosis. To avoid the lack of clarity the term has been replaced in the sentence. 

- We changed “pancreatic tissue” at line 543 to “pancreas”.

- Both solid (pancreas) and hollow (intestine) organs are cited in the text. The use of term “organ” seems correctly used throughout the text: “virtually all organ system may be involved”, “multi-organ immune-mediated condition”, “simultaneous or metachronous with other organ involvement”, “blood vessel walls in the affected organs”, “other organs (testicle, nervous system, skin)”, “virtually able to involve every organ system”. Available to review any confusing sentence.

Regarding the other points:

Like 281 and the rest of the text – the authors should ensure that the use of term “expression” is in relation to genes only.

RE: Thank you, the term has been changed.

Line 311 – please use refences in brackets. 

RE: Thank you. Done.

Line 572 – change to the “small intestine”

RE: Thank you, the term has been changed.

Line 493 - From the clinical point of view one of the most important symptoms related to both intestine and pancreas disorders is pain. Pathogenesis of pain is associated with changes in both peripheral and central nervous system. Noteworthy, intrapancreatic ganglia and enteric ganglia have the same sets of neurotransmitters/neuromodulators and some authors believe that intrapancreatic ganglia are prolongation of the enteric nervous system. In my opinion the innervation of the intestine and pancreas is in line with current review and should be (at least briefly) presented/discussed.

RE: Thank you for your insightful and inspiring comment. A brief comment about the modulation of pain sensitivity by immune cells and its implication in both IBD and chronic pancreatitis has been added in the article.

Reviewer 2 Report

There is a manuscript titled "A clinical and pathophysiological overview of intestinal and systemic diseases associated with pancreatic disorders: causality or casualty?" by Maria Cristina Conti Bellocchi and colleagues. Intestinal and pancreatic diseases, such as acute or chronic pancreatitis, autoimmune pancreatitis, or elevation of pancreatic enzymes, remain unclear. Associated with inflammatory bowel disease, acute pancreatitis (AP) is common, not only due to the use of drugs (such as Azathioprine), but also due to a variety of pathogenetic mechanisms, including chronic inflammation, alterations in microcirculation, gut permeability/motility, bacterial translocation, and activation of gut-associated lymphoid tissues. A number of factors are believed to contribute to celiac disease, including malnutrition, intestinal inflammation, disruption of enteric-mediated hormone secretion (cholecystokinin and secretin) and duodenal atrophy or papillary involvement, as well as AP. However, celiac disease has also been linked to exocrine pancreatic insufficiency and chronic asymptomatic pancreatic hyperenzymemia. Also, other systemic conditions (e.g., IgG4-related disease, sarcoidosis, vasculitis) may cause pancreatic inflammation. Regarding the present review, I would like to make a few comments.

-The introduction should summarize the state of the art in the field. There is a need for more information in this section regarding health measurements and their importance for clinical practice.

-There is no IBD explanation in the article. The AP should be introduced later in the article.

-References should appear in numerical order of appearance. Please refer to table 1.

-Check Table 2 line 2, Oh 2019.

-How did you decide to include celiac disease in your manuscript?

-What is the purpose of section 4 and why is it there?

-In my opinion, no objective of the definition is stated. The introduction of the subsections and the train of thought are necessary.

Author Response

Editor-in-Chief

Ms. Alexandra Negura

Assistant Editor

Biomedicines

Dear Editors,

we are grateful for the opportunity of sending a revised version of the manuscript entitled “A clinical and pathophysiological overview of intestinal and systemic diseases associated with pancreatic disorders: Causality or casualty?” for your consideration for publication in Biomedicines. We also thank the reviewers for their insightful comments that improved the quality of our paper.

Reviewer 2

There is a manuscript titled "A clinical and pathophysiological overview of intestinal and systemic diseases associated with pancreatic disorders: causality or casualty?" by Maria Cristina Conti Bellocchi and colleagues. Intestinal and pancreatic diseases, such as acute or chronic pancreatitis, autoimmune pancreatitis, or elevation of pancreatic enzymes, remain unclear. Associated with inflammatory bowel disease, acute pancreatitis (AP) is common, not only due to the use of drugs (such as Azathioprine), but also due to a variety of pathogenetic mechanisms, including chronic inflammation, alterations in microcirculation, gut permeability/motility, bacterial translocation, and activation of gut-associated lymphoid tissues. A number of factors are believed to contribute to celiac disease, including malnutrition, intestinal inflammation, disruption of enteric-mediated hormone secretion (cholecystokinin and secretin) and duodenal atrophy or papillary involvement, as well as AP. However, celiac disease has also been linked to exocrine pancreatic insufficiency and chronic asymptomatic pancreatic hyperenzymemia. Also, other systemic conditions (e.g., IgG4-related disease, sarcoidosis, vasculitis) may cause pancreatic inflammation. 

Regarding the present review, I would like to make a few comments.

-The introduction should summarize the state of the art in the field. There is a need for more information in this section regarding health measurements and their importance for clinical practice.

RE: Thank you for your suggestion. The introductive paragraph has been implemented and the importance of this topic in the clinical practice has been discussed.

-There is no IBD explanation in the article. The AP should be introduced later in the article.

RE: The definition of IBD and a brief introduction was added. Thank you. The AP is the most frequently observed pancreatic disorders in IBD patients and the most objectifiable, for both clinical impact and usually hospitalization need. For this reason, it is the first condition mentioned.

-References should appear in numerical order of appearance. Please refer to table 1. -Check Table 2 line 2, Oh 2019.

RE: The reference numerical order has been checked and corrected. Thank you

-How did you decide to include celiac disease in your manuscript?

RE: Thank you for your question. The celiac disease is included, since it is a chronic gastrointestinal disease with a reported increased risk of pancreatic disorders.

-What is the purpose of section 4 and why is it there?

RE: Knowing the co-occurrence of pancreatic and gastrointestinal involvement in several inflammatory and vasculitic diseases may be useful for the readers. Literature data and information about clinical onset and management are lacking. It seems in line with the current review.

Round 2

Reviewer 2 Report

I would like to thank the authors for taking the time to answer my questions regarding their previous version of the manuscript. The manuscript now reads well and all my comments have been addressed appropriately.